

SciPost Phys. Lect. Notes 40 (2022)

# Quantum enhanced metrology in the search for fundamental physical phenomena

Konrad W. Lehnert⋆

JILA, University of Colorado and National Institute of Standards and Technology,
Boulder, CO 80309-0440, USA

⋆ konrad.lehnert@jila.colorado.ed

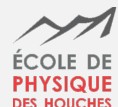

*Part of the Quantum Information Machines*
*Session 113 of the Les Houches School, July 2019*
*published in the Les Houches Lecture Notes Series*

## Abstract

These notes summarize lectures given at the 2019 Les Houches summer school on Quantum Information Machines. They describe and review an application of quantum metrology concepts to searches for ultralight dark matter. In particular, for ultralight dark matter that couples as a weak classical force to a laboratory harmonic oscillator, quantum squeezing benefits experiments in which the mass of the dark matter particle is unknown. This benefit is present even if the oscillatory dark matter signal is much more coherent than the harmonic oscillator that it couples to, as is the case for microwave frequency searches for dark matter axion particles.

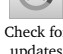

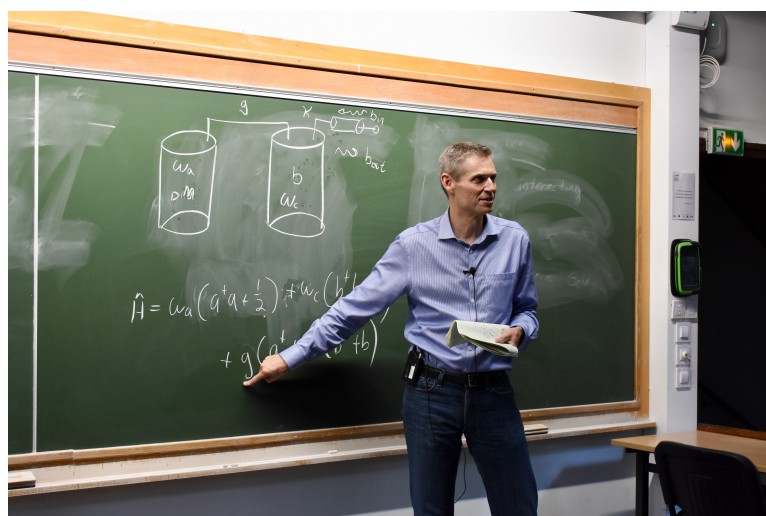

## 1  Introduction

These notes describe quantum enhanced measurement concepts that can be used in studies of fundamental physical phenomenon. I will focus on the particular case of immediate experimental relevance: ultralight or wave-like dark matter searches. As an abstract measurement problem it is quite similar to terrestrial detection of gravitational waves. Although the concept of using quantum enhanced sensing in searches of this type is more than 40 years old [1], deploying it to extend the reach of scientific instruments is a complex endeavor that required many years to become technically feasible. But there are now several recent results demonstrating quantum enhanced performance in a study of or search for fundamental phenomena. Notably, gravitational waves observatories now use quantum squeezing [2]. In addition, there are quantum enhanced searches for Lorentz invariance violation [3], hidden photons [4], and axionic dark matter [5].

For the case of dark matter, three recent trends have reinvigorated interest in this kind of quantum enhanced measurement. First, there is increasing activity searching for hypothetical dark matter particles that fall outside the domain of the so-called WIMP (Weakly Interacting Massive Particle) dark matter hypothesis. As yet, WIMP searches have not detected any dark matter [6], neither has there been direct [7] nor indirect [8] evidence for supersymmetric theories that underly the WIMP hypothesis, and thus interest is growing in other hypothetical particles. Second, the emergence of quantum computing technology using superconducting circuits has yielded a vast improvement in the ability to detect microwave signals generated inside of an ultralow temperature cryostat [9–12]. This ability is immediately applicable in searches for so-called axionic dark matter. Finally, the possibility of quantum enhanced sensing in axionic dark matter searches was overlooked partly because of confusion about the benefit

of quantum enhanced sensing [13].

## 2 Ultralight Dark Matter

It is widely accepted from astrophysical observations and the absence of laboratory interactions that dark matter is composed of an as yet unknown fundamental particle with the following properties: it 1.) is weakly interacting, 2.) is 'cold,' in that it is non-relativistic and, more strictly, gravitationally bound to our galaxy and to other clusters of galaxies, and 3.) has energy density $\rho_a \approx 0.4$ GeV/cm$^3$ [14, 15]. The first statement simply acknowledges that it must interact with ordinary matter weakly in order to have escaped laboratory detection. The second statement is a consequence of the virial theorem and assumes that gravitational interactions have established an equilibrium between ordinary matter and dark matter, thus implying a Maxwellian velocity distribution for dark matter with a characteristic velocity of $v \approx 300$ km/s $\approx c/1000$. The last statement is the value of missing mass density inferred from the visible matter and associated velocity distribution of our local cluster of galaxies.

These meager facts already have important implications for the quantum statistics of any fundamental, identical particles that would make up dark matter. If these hypothetical particles have rest-mass energy $m_a c^2$ less than about 10 eV, they must be bosons, and moreover, they are in a Bose condensed state. It is easy to estimate that for particles with $m_a c^2 < 10$ eV, the characteristic velocity implied by (2) yields a deBroglie wavelength $hc/[(m_a c^2)(v/c)] > 120\,\mu$m that, from (3) is larger than their average separation $1/\sqrt[3]{\rho_a/10\text{ eV}} \approx 30\,\mu$m. If the particles instead formed a degenerate fermi gas, their fermi velocity would be larger than the galactic virial velocity.

The best motivated of these ultralight dark matter theories is known as the quantum-chromodynamics (QCD) axion. It was originally proposed as a solution to the so-called "strong CP problem," which is one the famous unsatisfactory aspects of the standard model of particle physics. Specifically, given that the charge-parity (CP) symmetry is not preserved in nature, the fact that it is very well conserved in the strong nuclear sector seems an implausible accident. The hypothesis of Peccei and Quinn resolves this problem by positing a pseudo-scalar field that couples to quarks and undergoes a spontaneous symmetry-breaking phase transition [16]. Excitations of this field in its low energy phase, known as axions [17, 18], would have the appropriate properties to act as a source of dark matter [19–21].

An apparatus designed to search for such a light particle is quite different than most experiments that search for fundamental particles, where the hypothetical particle is usually much more massive. For example, hypothetical WIMP dark matter particles are believed to have rest mass energy between 10 GeV and 1 TeV. As such they would form a dilute gas of heavy particles, capable of delivering millions of electron-volts of energy in a very rare collision with a nucleus of similar rest-mass. Searches for this kind of dark matter thus use large volumes of very pure materials isolated from other sources of radioactivity [6].

In contrast, searches for ultra-light dark matter, with particle rest-mass energy less than one 1 eV, are better thought of as trying to detect an ever-present, large amplitude, but very weakly coupled *field*. A useful analogy is to think of detecting radio waves [22], which of course have a mathematically equivalent description in terms of particles with energy $\hbar\omega$, but are much more conveniently treated as a coherent field oscillating at $\omega$, which manifests as an oscillatory current flowing in an antenna. Experiments that search for this dark matter field often use a "table-top" laboratory apparatus, where the putative coupling of the dark matter causes a *parameter* in the Hamiltonian of the apparatus to oscillate at the Compton frequency of the dark matter $\omega_a = m_a c^2/\hbar$. This table top apparatus could be a microwave cavity [23], an inductor-capacitor resonant circuit [24], the collective spin of an ensemble of electron or

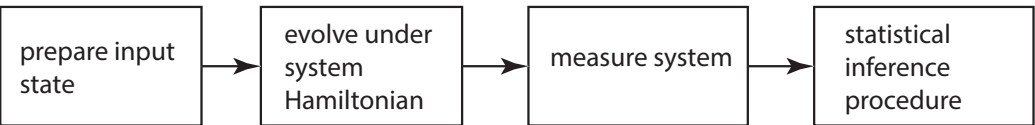

Figure 1: A flow chart for quantum metrology. To estimate unknown parameters of the dynamical system, 1.) prepare the system in some initial state, 2.) let it evolve for some time, 3.) make a measurement(s), and 4.) use classical statistical inference methods to estimate the unknown parameters.

nuclear spins [25], or a mechanical oscillator [26].

Although the quantum metrology concepts described here will be generally applicable to each of these cases, I will consider in detail the case of a search for axionic dark matter using a microwave cavity. Readers familiar with circuit quantum electrodynamics should grasp the essentials of the experimental apparatus even if the dark matter hypothesis is unfamiliar. At its heart, the apparatus is a mechanically tunable microwave cavity embedded in a large static magnetic field and cooled far below ambient temperature. The axionic dark matter hypothesis posits a modification to Maxwell's equations that yields a persistent oscillatory electric field parallel to the applied magnetic field at an unknown frequency. Should the frequency of the axion-derived signal be close enough to the cavity's resonance frequency, it can be detected by coupling the cavity to a transmission line and measuring the microwave field exiting the cavity. Given current experimental constraints, the cavity would emit (on average) much less than one axion-derived microwave photon per coherence time of the signal. As such, the quantum noise in a microwave field is an important limitation in determining whether the haloscope cavity evolves under a null hypothesis or if its Hamiltonian has been modified by the dark matter coupling.

## 3 Quantum Metrology

One of the main topics of quantum metrology is to design and evaluate strategies that estimate unknown parameters in a Hamiltonian [27]. The steps outlined in Fig.1 describe a measurement strategy. With a choice of an initial state of the Hamiltonian system with unknown parameters, an evolution time under that Hamiltonian, and a choice of measured observables, quantum theory yields a probability distribution of possible measurement outcomes, which depend on the unknown parameters in a known way. From this point, the problem is a matter of (classical) statistical inference of an unknown parameter from samples of a random process that depend on that parameter. The first three steps invite questions, such as: what is the best state to prepare the system in, for how long should it evolve in the system of interest, and what are the best quantities to measure? Of particular interest are quantum enhanced strategies, which out perform those that are described as classical or as subject to a quantum limit. The meaning of a 'classical' or 'quantum-limited' strategy depends on the particular context and should be explicitly described in any claim that a quantum enhanced method is superior.

A few remarks are in order before launching into a particular example.

1. To benefit from quantum enhanced strategies requires a level of technical perfection that is not achieved in very many physical systems. Measurement of and control over the system state has to be quite close to an ideal limit. It should be possible to prepare quantum states of high purity and the measurement uncertainty should be dominated by quantum projection noise. Although there are more systems for which that is true every year, they may not strongly couple to "fundamental physics" or hypothetical extensions

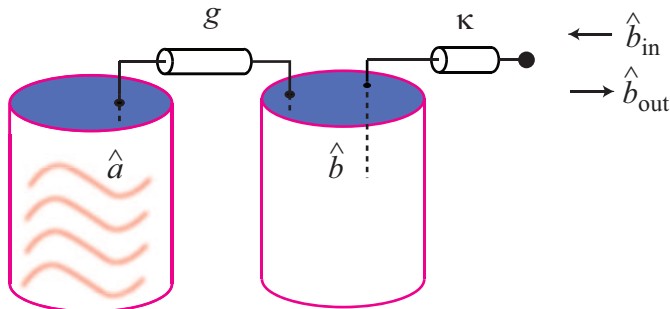

Figure 2: Toy model for ultralight dark matter coupling as a classical force. Two cavities coupled at rate $g$ will exchange energy. If $g$ is small compared to the energy decay rate $\kappa$ of the science cavity (right cavity: mode $\hat{b}$) and the dark matter mode is highly occupied (left: mode $\hat{a}$), the science cavity mode will evolve due to its coupling to the dark matter as if it were driven by an oscillatory force with negligible quantum noise compared to $\hat{b}_{\text{in}}$, the noise associated with its own decay and decoherence. Likewise the influence of the science cavity on the dark matter mode is also negligible.

to the standard model of particle physics. It is easier to manipulate the quantum state of small objects, while big objects presumably couple more strongly to fundamental physical phenomena that have as yet escaped laboratory detection.

2. The purpose of quantum enhanced measurements is to reduce the resources, essentially time and money, needed to reach some specified precision [27]. To improve measurement precision it is often possible to just wait longer, beating down the noise more. Likewise, one can also spend more money, maybe just by building multiple copies of the same experimental apparatus. Quantum enhanced methods become more relevant for mature concepts for which the resources needed to improve precision are both well understood and near a practical limit.

3. A measurement strategy involves a combination of quantum and classical statistical reasoning. Note that in the first three steps the laws of quantum mechanics are used to calculate a classical probability distribution. From the end of step three, everything is classical. Even so, the topic of parameter estimation from a known distribution is itself a gigantic (and historically contentious [28]) topic, which I will not attempt to cover in these notes.

## 3.1 Weak classical forces

The coupling of ultra-light dark matter to a laboratory apparatus can take different forms, such as a time varying resonance frequency or as a classical force. Here, I will consider the case that the putative dark matter coupling can be thought of as exerting a weak classical force on the quantum state of the experimental apparatus, as many of the prominent experimental efforts operate in this model [29]. What does that mean? Isn't the dark matter field to be detected a quantum field? Yes, but the associated quantum fluctuations may be hopelessly difficult to detect. Let's make a toy model to understand the point.

Imagine that the laboratory apparatus is a microwave cavity with resonance frequency $\omega_c$, weakly coupled to some dark matter with a portion of the field inside the microwave cavity. Modeling the dark matter mode inside the science cavity as if it were a second cavity

with resonance frequency $\omega_a$ prepared in a large amplitude coherent state and coupled to the science cavity at rate $g$, I can write down a Hamiltonian

$$\hat{H}/\hbar = \omega_a(\hat{a}^\dagger \hat{a} + 1/2) + \omega_c(\hat{b}^\dagger \hat{b} + 1/2) + g(\hat{a}^\dagger + \hat{a})(\hat{b}^\dagger + \hat{b})$$

and associated Heisenberg-Langevin equations of motion

$$\dot{\hat{a}} = -i\omega_a \hat{a} - ig(\hat{b}^\dagger + \hat{b}),$$
$$\dot{\hat{b}} = (-i\omega_c - \kappa/2)\hat{b} - ig(\hat{a}^\dagger + \hat{a}) + \sqrt{\kappa}\hat{b}_{\mathrm{in}},$$

where $\kappa$ is the total science cavity decay rate (modeled as if the cavity were coupled to a transmission line). The fluctuations associated with that dissipation are modeled by the noise operator $\hat{b}_{\mathrm{in}}$. These operators are described in more detail in section 6. To model the dark matter field in a large coherent state, I can use the displacement operator of quantum optics to transform the $\hat{a}$ mode as $\hat{a} \to \hat{D}^\dagger(\alpha)\hat{a}\hat{D}(\alpha) = \hat{a} + \alpha\mathbb{I}$, with $\hat{D}(\alpha) = \exp(\alpha^*\hat{a} - \hat{a}^\dagger\alpha)$. Now the average number of dark matter particles in the same volume as the science cavity is $\langle 0|\hat{D}^\dagger(\alpha)\hat{a}^\dagger \hat{a}\hat{D}(\alpha)|0\rangle = |\alpha|^2$, where $|0\rangle$ denotes the ground state of an oscillator.

Dark matter acts as a classical force in the limit of vanishingly small coupling $g \to 0$ and infinite amplitude $\alpha \to \infty$, such that their product is finite $0 < g|\alpha| < \infty$. To see this, solve the $\hat{a}$ equation of motion when $g = 0$ to find $\alpha(t) = \exp(-i\omega_a t)\alpha(0)$, and substitute into the $\hat{b}$ equation of motion

$$\dot{\hat{b}} = (-i\omega_c - \kappa/2)\hat{b} - ig(\alpha(0)e^{-i\omega_a t} + \alpha^*(0)e^{i\omega_a t}) + \sqrt{\kappa}\hat{b}_{\mathrm{in}}. \tag{1}$$

In this approximation, the quantum fluctuations of the dark matter field have no influence of the cavity's evolution. Those fluctuations arise from the commutation relations $[a, a^\dagger] = 1$, a tiny value compared to $|\alpha|^2$, where typical numbers in ultra-light dark matter searches range between $10^{10}$ and $10^{20}$. Furthermore, the cavity dissipation at rate $\kappa \gg g$ creates associated fluctuations $\sqrt{\kappa}b_{\mathrm{in}}$ much larger than those associated with the dark matter's quantum fluctuations. In other words, even though the dark matter field is assumed be very weakly coupled to the cavity, it is detectable in principle due its very large amplitude. But the science cavity's dissipation arising from microwave photons being converted into dark matter is negligibly small compared its intrinsic loss.

In fact, this example is closely analogous to the situation encountered in the terrestrial detection of gravitational waves, where very energetic pulses of gravitational waves pass the earth but deposit only a tiny amount of energy into a detector. Indeed, the notion of a classical force acting on quantum harmonic oscillator is the situation analyzed in the theory of gravitational wave detectors [1] and thinking about this problem was important motivation for the development of the theory of open quantum systems.

## 3.2 Theory of classical forces acting on a quantum oscillator

At this point it is convenient to introduce quadrature operators rotating at the science cavity frequency as

$$\hat{X} = (\hat{b}e^{i\omega_c t} + \hat{b}^\dagger e^{-i\omega_c t})/\sqrt{2}$$

and

$$\hat{Y} = -i(\hat{b}e^{i\omega_c t} - \hat{b}^\dagger e^{-i\omega_c t})/\sqrt{2},$$

with commutation relation $[\hat{X}, \hat{Y}] = i$. Substituting these into 1 yields equations of motion for these rotating quadrature amplitudes

$$\dot{\hat{X}} = -\frac{\kappa}{2}\hat{X} + F_0 \sin(\Delta t + \phi) + \sqrt{\kappa}\hat{x}_{\mathrm{in}} \tag{2}$$

and

$$\dot{\hat{Y}} = -\frac{\kappa}{2}\hat{Y} - F_0\cos(\Delta t + \phi) + \sqrt{\kappa}\hat{y}_{\text{in}}, \tag{3}$$

where $F_0 = \sqrt{2}g|\alpha|$, $\phi = \arg(\alpha)$, and $\Delta = \omega_c - \omega_a$. The noise terms $\hat{x}_{\text{in}}$ and $\hat{y}_{\text{in}}$ are defined analogously to $\hat{X}$ and $\hat{Y}$.

The utility of these equations is that they provide a convenient geometrical picture of the state of the science cavity and its evolution. First, in the absence of an external force $\alpha = 0$ and dissipation $\kappa = 0$ this representation is stationary; i.e., the Hamiltonian evolution of the harmonic oscillator is absorbed into the definition of the rotating quadratures. If $\alpha \neq 0$, evolution under a classical force will act only to displace the state in phase space. (Notice that Eqs. 2 and 3 are complete equations of motion for the system; these equations are not coupled to higher moments of $\hat{X}$ or $\hat{Y}$.). Finally, the state of the harmonic oscillator is compactly represented using the Wigner distribution function $W(X, Y)$, which shares many features of a phase-space distribution in a classical statistical-mechanics analysis of a harmonic oscillator. If one performs only *linear* measurements of $\hat{X}$ *or* $\hat{Y}$, all statistical properties for the outcomes of such a measurement are calculated by treating the Wigner function as if it were a properly normalized probability distribution function [30]. For example, the probability density function $P(X) = \int_{-\infty}^{\infty} W(X, Y)\,dY$ to measure a particular value $X$ of the $\hat{X}$ observable is found by integrating over the unmeasured phase space coordinate and thus the expectation value for $n^{\text{th}}$ moment of $X$ is $\langle X^n \rangle = \int_{-\infty}^{\infty} X^n W(X, Y)\,dY\,dX$.

In spite of these properties, $W(X, Y)$ is definitely not a classical probability distribution function[1], and the distinction is crucial to defining a quantum limited measurement of a classical force. Consider the Wigner function of the harmonic oscillator ground state $W_0 = (1/\pi)\exp[-(X^2 + Y^2)]$, from which it is easy to calculate the variance in a measurement of $\hat{X}$ or $\hat{Y}$ as $\text{Var}(X) \equiv \langle X^2 \rangle - \langle X \rangle^2 = 1/2 = \text{Var}(Y)$. Note that the calculation using the Wigner function is consistent with the commutation relation $[\hat{X}, \hat{Y}] = i$, with associated uncertainty relation $\text{Var}(X)\text{Var}(Y) \geq (1/4)$, and with the straightforward evaluation of $\langle 0|\hat{X}^2|0\rangle$. But a simultaneous measurement of both non-commuting observables has a probability distribution function $P_0(X, Y) = (1/(2\pi))\exp[-(X^2 + Y^2)/2]$, with twice the variance $\text{Var}(X) = \text{Var}(Y) = 1$ [30]. The additional unit of quantum noise is usually attributed to some process of measurement and any real measurement procedure could introduce more. But the added unit of quantum noise is inevitable and any measurement of both quadratures of a harmonic oscillator which adds exactly one unit of quantum noise is said, in this context, to be quantum-limited [31].

To understand the implication for sensing a classical force, I write an effective Hamiltonian whose equations of motion are the $\kappa = 0$ limit of Eqs. 2 and 3

$$\hat{H}/\hbar = \hat{X}F_X(t, \phi) + \hat{Y}F_Y(t, \phi), \tag{4}$$

with $F_X = F_0\cos(\Delta t + \phi)$ and $F_Y = F_0\sin(\Delta t + \phi)$. If the goal is to measure the values of $F_X$ and $F_Y$ precisely, it is clear that the force can be inferred from measurements of $\hat{X}$ and $\hat{Y}$. But quantum fluctuations will add noise to this inference.

Can this noise be overcome? Localizing $X$ and $Y$ to a point in phase space is not consistent with quantum physics. But examining Eq. 4, we see that the quantities we want to know are just numbers. They are simply *parameters* that enter the Hamiltonian. There is no principle of quantum mechanics that precludes arbitrarily precise knowledge of $F_X$ or $F_Y$. For that matter, both can be measured simultaneously with (in principle) unbounded precision for a particular measurement time. But the precision will be bounded in a strategy that seems

---

[1]This is most strikingly seen from that fact that many quantum states have negative regions in their Wigner functions. In addition, measuring $\hat{X}$ on an ensemble of identically prepared states and calculating $\langle X^n \rangle$ is not equivalent to building a meter that directly senses $\hat{X}^n$ and finding the expectation value of those measurements.

(erroneously) to be optimal. The strategy that defines the quantum limit for this kind of force measurement is to prepare the harmonic oscillator in its ground state, allow it to evolve for the measurement time, and then measure both $\hat{X}$ and $\hat{Y}$ with the minimum one unit of quantum noise (sometimes referred to as one half-photon of noise) in each quadrature. A measurement strategy that achieves better precision for the same measurement time, or more practically, the same precision in a shorter time is said to beat the quantum limit or the standard quantum limit.

Similarly, one could forgo the ability to measure $F_X$, and instead just attempt to learn the value of $F_Y$. In that case, the same initial ground state could be prepared, the system allowed to evolve, and only $\hat{X}$ measured. Even though the measurement can in principle add no noise, the precision of the force inference is still "quantum-limited" by the noise present in the $X$-quadrature of the initial ground state.

For both cases, preparing the system in its ground state yields a quantum limit. I will refer to both cases as operating at the coherent state limit (CSL) and measurement strategies that outperform this strategy beat the CSL [32] [2].

# 4 Quantum Enhanced Sensing with Squeezed States

## 4.1 Single mode squeezing

Returning to Eq. 4, let's think about how $F_Y = F_0 \sin(\Delta t + \phi)$ can be estimated. Imagine, for the moment, that I know $\omega_a$ such that I can build my laboratory apparatus to be resonant with the force ($\Delta = 0$), and that I can prepare that system in its ground state with $\langle X \rangle = 0$ and $\mathrm{Var}(X) = 1/2$. Evidently, if I allow the system to evolve for a time $t_s$, $\hat{X}(t_s) = \hat{X}(0) + F_Y t_s \mathbb{I}$. Now if I measure observable $\hat{X}$ and get the outcome $X$, the estimate for the true value of $F_Y$ is $\tilde{F}_Y = X/t_s$. Notice that the estimate is unbiased in the sense that $\langle F_Y - \tilde{F}_Y \rangle = 0$ and it has CSL variance $\mathrm{Var}(\tilde{F}_Y) = 1/2t_s^2$.

The noise in the CSL estimate of $F_Y$ comes from the fact that a coherent state is not an eigenstate of the measurement operator $\hat{X}$. A seemingly straightforward way to eliminate this noise completely is to prepare the oscillator in an eigenstate of the $\hat{X}$, rather than the ground state.

Such a state is unphysical, but a state squeezed along the $X$ quadrature approaches it in the limit of large squeezing. Squeezing is a unitary transformation of a harmonic oscillator state with operator representation in $\hat{b}$ and $\hat{b}^\dagger$

$$\hat{S}_X(r) = \exp\left[\frac{1}{2}(r\hat{b}^2 - r\hat{b}^{\dagger 2})\right],$$

with $r$ the real valued squeezing parameter. More useful for our purpose are properties

$$\hat{S}_X^\dagger \hat{X} \hat{S}_X = e^{-r}\hat{X} \equiv \hat{X}/\sqrt{G},$$
$$\hat{S}_X^\dagger \hat{Y} \hat{S}_X = e^{r}\hat{Y} = \sqrt{G}\hat{Y},$$

where $G = e^{2r}$ is the squeezing factor, also known (a little counterintuitively) as the squeezing gain.

From these relations, it is clear that the squeezing operation is a rescaling of the phase-space coordinates that reduces the quantum noise in the $X$ quadrature. If this squeezing operation is performed on the ground state, the resulting state is represented by a Wigner

---

[2]Coherent states include the ground state and any state whose Wigner function can be described as a ground state Wigner function displaced from the origin in phase-space.

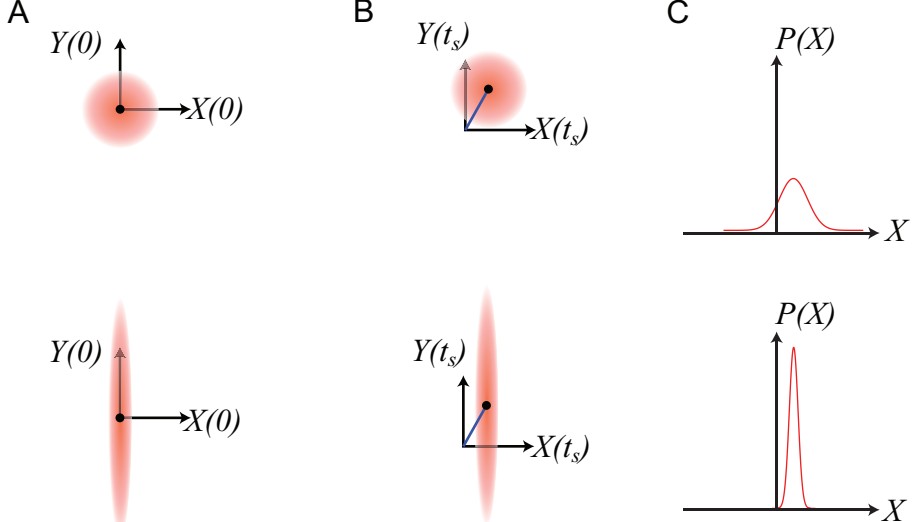

Figure 3: A depiction of single mode squeezing enhanced sensitivity. The two rows compare the coherent state strategy (top row) to a squeezed state strategy (bottom row). A.) First the system is prepared in either the ground state or a squeezed state shown as Wigner density plots. B.) Under evolution for a time $t_s$ the states are displaced in phase space by the unknown classical force. C. Plotting the resulting probability density function shows that a squeezed state yields a measurement of $X$ with less uncertainty for the $X$ component of the displacement, thereby reducing the uncertainty in one quadrature of the classical force.

function $W_{\mathrm{SMGS}} = (1/\pi)\exp[-(GX^2 + Y^2/G)]$. The $X$ variance of a squeezed vacuum state is $\langle 0|\mathrm{Var}(\hat{S}_X^\dagger \hat{X}\hat{S}_X)|0\rangle = e^{-2r}/2 = 1/2G$ is indeed squeezed, while the $Y$ variance is antisqueezed to $e^{2r}/2 = G/2$.

As shown in Fig. 3, let's amend the measurement strategy for $F_Y$ from the CSL strategy, so that 1.) the cavity is first prepared in a squeezed state, 2.) allowed to evolve for time $t_s$, 3.) then $\hat{X}$ is measured with outcome $X$, and finally 4.) $F_Y$ is estimated as $\tilde{F}_Y = X/t_s$. Beginning in a squeezed vacuum state rather than vacuum has definitely beaten the CSL because $\mathrm{Var}(\tilde{F}_Y) = 1/(2Gt_s^2)$.

## 4.2 Two mode squeezing

You might have already noticed that this reduction in the variance of $\tilde{F}_Y$ has been balanced by a proportionate increase in the variance that would have been achieved in a measurement of $F_X$. That is, if the system were prepared in the same squeezed state but instead only $Y$ was measured, the associated estimate would have greatly increased variance $\mathrm{Var}(\tilde{F}_X) = G/2t_s^2$. It is tempting *but wrong* to view this as some kind quantum morality tale, in which the benefit of squeezing in the measurement of a parameter coupled to $Y$ is balanced by an associated penalty in measuring a parameter coupled to the canonically conjugate observable $X$.

The measurement strategy for overcoming this apparent compromise is a beautiful example of using an entangled ancillary system to enhance sensitivity [33]. Imagine that I introduce a second oscillatory system (a second cavity in our example) whose quantum state can be manipulated, but can be otherwise much simpler than the science cavity as it need not couple to the dark matter field. Denote the state of the science cavity with phase space coordinates $X_1$ and $Y_1$, and the ancillary cavity $X_2$ and $Y_2$. Particular joint observables $\hat{Q} = \hat{X}_1 - \hat{X}_2$ and $\hat{P} = \hat{Y}_1 + \hat{Y}_2$, commute with each other $[Q, P] = 0$. (The two other observables orthogonal to

respectively $\hat{Q}$ and $\hat{P}$, are $\hat{R} = \hat{X}_1 + \hat{X}_2$ and $\hat{S} = \hat{Y}_1 - \hat{Y}_2$.). It is therefore possible to prepare the joint system in a simultaneous eigenstate of both $\hat{Q}$ and $\hat{P}$, known as an Einstein, Padolsky, and Rosen (EPR) state [34].

An EPR state also has an unphysical character but, just as for a single-mode squeezed state, a two-mode squeezed state approaches an EPR state in the limit of infinite squeezing. The two mode squeezing operator is

$$\hat{T}_Q = \exp\left[\frac{1}{2}(\hat{b}_1\hat{b}_2 r^* - \hat{b}_1^\dagger \hat{b}_2^\dagger r)\right],$$

where again $r$ is the squeezing parameter, here chosen to be real to select $Q$ and $P$ as the squeezed coordinates. The operator transformations are

$$\hat{T}_Q^\dagger \hat{Q} \hat{T}_Q = e^{-r}\hat{Q}$$

and

$$\hat{T}_Q^\dagger \hat{P} \hat{T}_Q = e^{-r}\hat{P}.$$

Operators $\hat{R}$ and $\hat{S}$ are proportionally anti-squeezed. From their definitions, the variance of each of these operators is, for example, $\langle 00|\text{Var}(\hat{Q})|00\rangle = 1$, when both modes are in their ground state (or in any coherent state).

Now imagine a protocol in which both modes begin in their ground state, are transformed by a two-mode squeezing operation, allowed to evolve for time $t_s$, and then both $\hat{Q}$ and $\hat{P}$ are measured.

$$\hat{Q}(t_s) = \hat{T}_Q^\dagger \hat{Q}(0)\hat{T}_Q + F_Y t_s \mathbb{I},$$
$$\hat{P}(t_s) = \hat{T}_Q^\dagger \hat{P}(0)\hat{T}_Q - F_X t_s \mathbb{I}.$$

From the measurement of $\hat{Q}$ we get one outcome $Q$ and likewise $P$. Defining our estimators

$$\tilde{F}_Y = Q/t_s$$

and

$$\tilde{F}_X = -P/t_s,$$

we find variances

$$\text{Var}(\tilde{F}_Y) = \text{Var}(\tilde{F}_X) = \frac{e^{-2r}}{t_s^2}.$$

Thus, both force components can be measured with arbitrary precision in the limit of arbitrarily large squeezing.

Although it is important to understand that there is no principle of quantum mechanics that precludes a simultaneous, noiseless estimate of both quadratures of a classical force, this ability is not essential for the situation considered here. In fact, in a microwave frequency axion dark matter search, there is no clear benefit to using a two-mode squeezed state rather than a single-mode squeezed state with equivalent squeezing factor $G$. Essentially, the doubling of the signal power from measuring both quadratures is compensated by a doubling of the squeezed quantum noise from measuring two modes rather than one (see Appendix C in [13]). Furthermore the technical complexity of implementing a two-mode receiver is likely to be greater than a single-mode squeezer. As such, in the remaining notes I consider just single-mode squeezing.

# 5 Loss and Decoherence

As presented in Sec. 4, squeezing seems to be a miraculous tool to overcome quantum noise. But when considering the inevitable loss and associated decoherence of the microwave cavity, the benefit seems to disappear. Understanding that squeezing indeed provides a real benefit to a dark matter search was part of the confusion that needed to be resolved before quantum enhanced measurement techniques were adopted.

Before calculating in detail, let's first get some intuition looking at Eqs. 2 and 3. When $\kappa \neq 0$ but the cavity is still resonant with the dark matter signal ($\Delta = 0$), the steady state limit is

$$\langle X \rangle = 2F_Y/\kappa \,.$$

The estimate is then $\tilde{F}_Y = X\kappa/2$. If $\hat{x}_{\text{in}}$ and $\hat{y}_{\text{in}}$ are noise operators associated with loss to a zero temperature environment, the oscillator will be in a coherent state, so that the estimator's variance is $\text{Var}(F_Y) = \kappa^2/8$. Notice that this estimate does not improve with time anymore. When there is no dissipation, a classical force applied on resonance displaces a oscillator's state by an amount that grows linearly with time, and without bound. But with dissipation, the displacement caused by an on resonance force reaches a steady state and waiting longer does not yield an increase in the displacement signal. Furthermore the input noise operators in this linear system behave as noisy diffusive terms that act to increase the variance of the quadratures at the same rate that it decays from the dissipative terms in Eqs. 2 and 3. In this Heisenberg-Langevin picture, the oscillator's quantum noise appears to be sourced by these noise operators.

It is certainly possible to measure the $X$-quadrature repeatedly in order to get many realizations of the noise process $\hat{x}_{\text{in}}$, but by solving Eq. 2

$$\hat{X}(t) = \hat{X}(0)e^{\frac{-\kappa t}{2}} + \int_0^t \mathrm{d}t' e^{\frac{-\kappa(t-t')}{2}} \left( \sqrt{\kappa}\hat{x}_{\text{in}}(t') + F_Y(t') \right), \tag{5}$$

it is clear that these measurements will be correlated for a characteristic time $2/\kappa$. The number of *independent* measurements in a time $t_s$ is then $N = (\kappa/2)t_s$. For $N$ independent measurements of $F_Y$ the variance should reduce as $1/N$ [35]. Putting this together I expect

$$\text{Var}(F_Y)(t_s) = \kappa/4t_s \,.$$

Notice that the variance in the estimate only reduces as the inverse measurement time, not its square. Equation 5 also shows that in the $t_s \gg 1/\kappa$ limit, preparing the system initially in a squeezed state does not improve the situation as an initially reduced variance returns to the ground state at a characteristic rate $\kappa$.

Rather than imaging a world without decoherence, I can think about estimating $F_Y$ on a timescale short or long compared to $1/\kappa$. In particular, if $F_Y$ is a quantity that varies with time and I want to track it, I should sample its value more rapidly than its coherence time. For example the phase $\phi$ could vary with characteristic coherence time $\tau_0$ such that $F_Y$ would have an autocorrelation function

$$\langle F_Y(t)F_Y(t+\tau)\rangle_t = (F_0^2/2)e^{-\tau/\tau_0}\cos(\Delta\tau) \,.$$

If $\tau_0 \ll 1/\kappa$, quantum metrology is clearly beneficial.

For the particular case of microwave frequency axionic dark matter the situation is just the opposite [36, 37]. The cavity used to detect axionic dark matter must reside in a large static magnetic field incompatible with a high-$Q$ superconducting cavity [38]. As such it is made of

copper and its quality factor is only about 10,000 to 100,000. In contrast the coherence of the axion signal itself can be estimated from the galactic virial velocity $v/c \approx 0.001$. In the frame of the detector the frequency of the axion signal will vary because of the spread in kinetic energy by a characteristic amount $(1/2)m_a v^2/\hbar$. The frequency will have a minimum value $m_a c^2/\hbar$ and a fractional linewidth of $\sim (v/c)^2 \approx 10^{-6}$. Thus the axion signal itself is believed to be about 10 to 100 times more coherent than the cavity used to detect it.

If one knew the axion frequency, squeezing would indeed be of no benefit. Precisely because the axion frequency is unknown, the spectral width over which the cavity is sensitive is crucial. Axionic dark matter experiments tune a resonant microwave cavity through frequency space searching for an axion signal. For the same sensitivity to an axion signal, a measurement procedure that yields a wider frequency range of sensitivity—a wider bandwidth—is beneficial because it allows a more rapid search through frequency. Squeezing benefits an axion search by preserving the on-resonance sensitivity over a wider bandwidth than the quantum-limited strategy.

Qualitatively, the measurement bandwidth is $1/t_s$. In the absence of squeezing, measurements should be made at the rate $\kappa$, and the measurement bandwidth is itself $\kappa$, with variance $\text{Var}(F_Y) = \kappa^2/4$. By measuring much more quickly, the bandwidth is proportionally increased and the sensitivity to the axion reduced. But in this fast measurement strategy, squeezing can be employed to recover the lost sensitivity.

# 6 Quantum Optics Model for an Axion Search

To go beyond these qualitative arguments, I adapt the model introduced in Fig. 1 to create a model (Fig. 4) for a microwave-frequency axion measurement apparatus whose behavior can be calculated using the formalism of the input-output theory of quantum optics and for which the benefit of one-mode squeezing can be calculated in detail. The science cavity mode now exchanges energy with three distinct environments (ports): a real measurement port engineered to couple the cavity to an amplifier through a transmission line, a fictitious port modeling the cavity's internal loss, and a hypothetical port associated with the putative axion-photon interaction. The dark matter cavity in Fig. 1 is now replaced by an oscillatory signal generator because the axion coupling to the cavity is always too weak to deplete the local density of dark matter. The amplitude of this generator and its coupling to the science cavity can be expressed in terms of the dark matter density and the coupling of axion models to electromagnetism [13, 39, 40]. Rather than assume that the $X$-quadrature can be measured directly, this model is closer to the experimental reality in which the field exiting the science cavity through its measurement port is then measured by linear amplification and the $X$-quadrature inferred from the time-continuous amplifier output. Likewise, rather than imagining that the cavity field is squeezed directly, instead the input field can be continuously squeezed and transported to the cavity. (This model can be adapted to describe a realistic implementation of the two-mode scheme presented in Sec. 4.2 as described in Sec. II of Ref. [41].) For continuous, linear, single-quadrature amplification and squeezing of microwave frequency signals, Josephson parametric amplifiers (JPAs) are currently the best available technology [9, 42–44].

The transmission lines and resistors in the figure are a way to schematically represent the interaction of the cavity with its three environments. In the rotating frame of the science cavity, the Heisenberg-Langevin equation of motion for the cavity field is

$$\dot{\hat{b}} = -\frac{\kappa}{2}\hat{b}(t) + \sum_j \sqrt{\kappa_j}\hat{b}_{\text{in},j}, \tag{6}$$

where $\hat{b}_{\text{in},j} \in \{\hat{b}_{\text{in,m}}, \hat{b}_{\text{in,l}}, \hat{b}_{\text{in,ax}}\}$ are the annihilation operators (in the same rotating frame) of

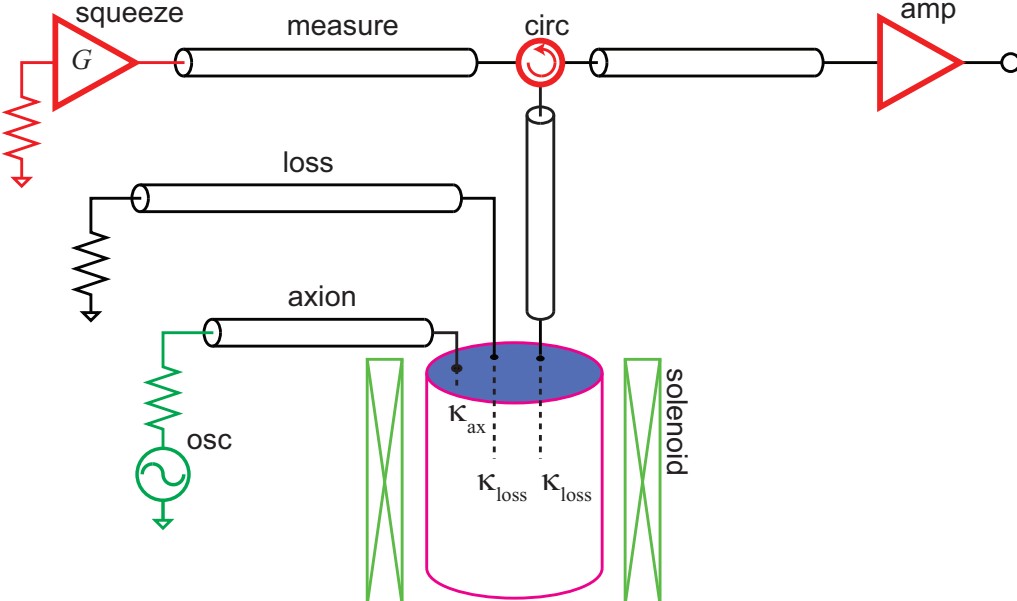

Figure 4: Diagram of an axion dark matter search apparatus. A tunable microwave cavity is cooled far below ambient temperature and embedded in large magnetic field (solenoid), which—in the presence of an axion field—generates a feeble electric field oscillating a frequency $m_a c^2/\hbar$ modeled as if it were caused by a microwave oscillator (osc) with a large amplitude but weak coupling. The fluctuation and dissipation of the cavity are modeled as arising from three coaxial cables that protrude into the cavity mode. These ports extract energy and deliver noisy fields from the quantum-Nyquist noise of the resistors that terminate the cables. The measurement port is in fact a coaxial cable whose coupling $\kappa_{\mathrm{m}}$ can be adjusted while the cavity is cold. It is distinct because the source of its fluctuations are experimentally accessible and its incident fields can be squeezed (squeeze) and separated from its outgoing fields using a microwave circulator (circ). The outgoing fields are then measured by an amplifier (amp).

the modes of the environment with commutation relations $[\hat{b}_{\mathrm{in},j}(t), \hat{b}_{\mathrm{in},k}^\dagger(t')] = \delta(t-t')\delta_{jk}$. In the model, they are the input fields in the transmission lines incident on the measurement, loss, and axion port respectively and are sourced by the quantum Johnson-Nyquist noise of the resistors terminating the lines and the oscillatory dark matter signal. Likewise $\kappa_j \in \{\kappa_{\mathrm{m}}, \kappa_{\mathrm{loss}}, \kappa_{\mathrm{ax}}\}$ are the rates that the cavity energy decays to the three ports, and $\kappa = \sum_j \kappa_j$. The output fields from the cavity are related to the incident fields and to the cavity mode according to input-output relations [45] as

$$\hat{b}_{\mathrm{out},j} = \hat{b}_{\mathrm{in},j} - \sqrt{\kappa_j}\hat{b}(t). \tag{7}$$

These linear equations of motion and input-output relations can be solved in the Fourier domain as $\hat{b}_{\mathrm{out},j}(\omega) = \sum_k \chi_{jk}\hat{b}_{\mathrm{in},k}(\omega)$ where,

$$\chi_{jk}(\omega) = \frac{-\sqrt{\kappa_j}\sqrt{\kappa_k} + (\kappa/2 + i\omega)\delta_{jk}}{(\kappa/2 + i\omega)}, \tag{8}$$

are the elements of linear susceptibility. Finally, the noise in the output fields is determined by the noise in the input fields and $\chi_{jk}$. The input noise power spectrum (in units of photons per second) is characterized by

$$\langle \hat{b}_{\mathrm{in},j}^\dagger(-\omega')\hat{b}_{\mathrm{in},k}(\omega)\rangle = 2\pi\bar{n}_j(\omega')\delta(\omega - \omega')\delta_{jk}, \tag{9}$$

where $\bar{n}_j(\omega)$ is the spectral density of the noise incident on port $j$, in units of photons. If the measurement and loss ports model environments that are in thermal equilibrium at temperature $T$, the thermal average number of photons in a mode of frequency $\omega_c + \omega$, is approximated by its value at cavity resonance $\bar{n}_m(0) = \bar{n}_{loss}(0) = \bar{n}_T = [\exp(\hbar\omega_c / k_B T) - 1]^{-1}$ [45].

When transforming to the quadrature basis as $\hat{x}_{in,j}(\omega) = (\hat{b}_{in,j}(\omega) + \hat{b}^{\dagger}_{in,j}(-\omega))/\sqrt{2}$ and $\hat{y}_{in,j}(\omega) = (\hat{b}_{in,j}(\omega) - \hat{b}^{\dagger}_{in,j}(-\omega))/(\sqrt{2}i)$, the susceptibility conveniently keeps the same form such that $\hat{x}_{out,j} = \sum_k \chi_{jk}(\omega)\hat{x}_{in,k}(\omega)$, $\hat{y}_{out,j} = \sum_k \chi_{jk}(\omega)\hat{y}_{in,k}(\omega)$, and the $x$ and $y$ quadratures are uncoupled. Just as for the toy model in Sec. 4.1, the effect of squeezing is simple to represent on quadrature operators $\hat{x}_{m,in}(\omega) \to \hat{x}_{m,in}(\omega)/\sqrt{G(\omega)}$ and $\hat{y}_{m,in}(\omega) \to \hat{y}_{m,in}(\omega)\sqrt{G(\omega)}$.

## 6.1 Quantum noise in an axion search apparatus

With these definitions and solutions established, it easy to write down the power spectrum of the noise in a measurement of the $x$ quadrature field component that exits the cavity at the measurement port as $\langle \hat{x}_{m,out}(\omega)\hat{x}_{m,out}(\omega') \rangle = 2\pi S_{x,out}(\omega)\delta(\omega - \omega')$ where

$$S_{x,out}(\omega) = |\chi_{mm}|^2 \frac{(\bar{n}_T + \frac{1}{2})}{G} + |\chi_{ml}|^2 \left(\bar{n}_T + \frac{1}{2}\right) + |\chi_{ma}|^2 \left(n_{ax} + \frac{1}{2}\right), \qquad (10)$$

and $n_{ax}(\omega)$ is the number spectral density of the axion field.

If the axion exists and makes up dark matter, $n_{ax}(\omega)$ is a sharply peaked function centered near $\omega_a - \omega_c$ (in the cavity's rotating frame) with a characteristic width $\delta_a \approx \omega_a/10^6$ (which, again, is currently narrower than an axion cavity linewidth $\kappa \approx \omega_c/10^4$). The relevant quantity is then $\sigma(\omega)$ the fractional change in $S_{x,out}(\omega)$ between the null hypothesis $n_{ax}(\omega) = 0$ and the axion dark matter hypothesis $n_{ax}(\omega) > 0$.

$$\begin{aligned} \sigma(\omega) &\equiv \frac{S_{x,out}(\omega) - S_{x,out}(\omega)|_{n_{ax}=0}}{S_{x,out}(\omega)} \\ &= \frac{|\chi_{ma}|^2 (n_{ax})}{S_{x,out}}. \end{aligned} \qquad (11)$$

This ratio of spectral densities will determine how long one must average to resolve excess axion power in each frequency band of width $\delta_a$. In the physically accessible limit of axion to photon conversion $\kappa_{ax} \ll \kappa_{loss} \sim \kappa_m$ and using Eq. 8 this expression simplifies to

$$\sigma(\omega) = \left(\frac{n_{ax}}{\bar{n}_T + \frac{1}{2}}\right)\left(\frac{\kappa_{ax}\kappa_m}{\kappa_m \kappa_{loss} + \beta(\omega)/G}\right), \qquad (12)$$

where $\beta(\omega) = [((\kappa_m - \kappa_{loss})/2)^2 + \omega^2]$.

## 6.2 Accelerating an axion search with quantum squeezing

Examining this expression in several limits reveals the behavior intuited in Sec. 5. First, if the axion frequency were known, squeezing would provide no benefit. To see this, imagine fixing $\kappa_{loss}$ at the smallest achievable value for a copper cavity [38] and maximize the on resonance sensitivity $\sigma(\omega = 0)$ over $\kappa_m$. The best sensitivity occurs at critical coupling where $\kappa_m = \kappa_{loss}$, for any value of the squeezer gain $G$, where $\sigma(0) = [n_{ax}/(\bar{n}_T + 1/2)](\kappa_{ax}/\kappa_{loss})$. This on-resonance and critically coupled sensitivity is the technically-limited axion sensitivity. It can be improved by advances that increase $n_{ax}$ or $\kappa_{ax}$, or decrease $\kappa_{loss}$ or $\bar{n}_T$, but not by squeezing. Second, squeezing is useful in the search for an axion of unknown frequency because it increases the bandwidth. Consider that in the critically coupled case and in the

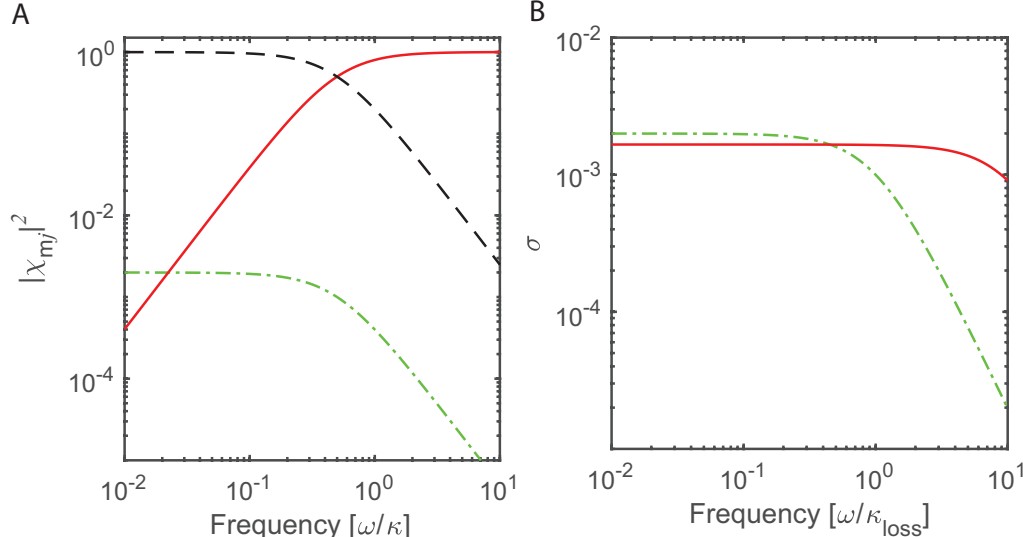

Figure 5: Influence of squeezing on the measurement bandwidth. A.) The square magnitude of susceptibility matrix elements: $|\chi_{mm}|^2$ (red solid), $|\chi_{ml}|^2$ (black dashed), $|\chi_{ma}|^2$ (green dashed dotted) are plotted as a function of Fourier frequency ($\omega$) detuned from the cavity's resonance, with $\kappa_{ax} = \kappa/1000$ and $\kappa_m = \kappa_{loss}$. Even this value of $\kappa_{ax} \ll \kappa$ is implausibly large, but chosen so that all of the elements can be plotted on the same logarithmic scale. B. Axion sensitivity: The sensitivity $\sigma(\omega)$ is plotted for two cases: critically coupled $\kappa_m = \kappa_{loss}$ with no squeezing G= 1 (green dash dotted) and overcoupled $\kappa_m = 10\kappa_{loss}$ with squeezing G=10 (red solid).

absence of squeezing $G = 1$, the frequency band over which the maximum sensitivity is no more than halved is $\kappa_{loss}$, defining the bandwidth. But this bandwidth limitation is dramatically improved by squeezing, and in the limit $G \to \infty$, the maximum sensitivity is achieved at any frequency and for any value $\kappa_m$. Finally, squeezing is beneficial even if the environment is hot $\bar{n}_T \gg 1$, which is the case in axion dark matter experiments that search in the kilohertz to megahertz frequency range [24]. In that case, thermal noise rather than quantum noise limits the sensitivity. But thermal noise can be squeezed in just the same way that quantum noise can[3].

These results can be understood by considering the ways in which the loss and measurement ports are inequivalent (Fig. 5A). Even without a calculation, it is clear that noise entering the cavity through the axion port and loss port will be altered identically by the cavity response on the way to the amplifier. The ratio $|\chi_{ma}|^2/|\chi_{ml}|^2$ is independent of $\omega$ and $\kappa_m$, and if the loss port noise were the only source of undesired fluctuations the axion search apparatus would have a bandwidth much larger than $\kappa$. In contrast, noise incident on the cavity through the measurement port may be promptly reflected at that port and bypass the cavity to reach amplifier. But unlike the loss-port noise, these fluctuation arise from a source whose quantum state can be controlled, namely a resistor on one port of the circulator (Fig. 4). As such, one quadrature of any noise reaching the amplifier from this source could have been reduced by squeezing.

Viewed this way, the search apparatus with squeezing is a kind of back-action evading measurement. The measurement port coupling $\kappa_m$ controls both the rate at the which the cavity's state is measured and the rate at which it is disturbed by that measurement, as seen directly from the input-output relations (Eqs. 6 and 7). By increasing $\kappa_m$ while squeezing one quadra-

---

[3]Although in the limit $\bar{n}_T \gg 1$, a quantum limited amplifier alone can dramatically reduce the variance from an initial thermal state to a post-measurement coherent state without requiring any squeezing resource [46].

ture of the input field, the back-action associated with more rapid measurement is deposited in the unmeasured $y$-quadrature. For finite $\kappa_\mathrm{m}$ and $G$, the net effect of squeezing is a small reduction in the on-resonance sensitivity coupled with a large increase in bandwidth(Fig. 5B).

To understand the compromise between wider bandwidth but reduced $\sigma(0)$, I find the rate at which the cavity can be tuned in the search for an axion signal by finding the averaging time $t_\mathrm{av}$ needed to resolve an axion signal near cavity resonance, and the characteristic size of a step in the cavity's resonance frequency. To find $t_\mathrm{av}$, it is helpful to interpret the quantity $\sigma(\omega)$ as the fractional precision with which the variance $S_{x,\mathrm{out}}(\omega)$ must be determined in order to detect an axion with unit signal to noise ratio. The statistical uncertainty in determining $S_{x,\mathrm{out}}(\omega)$ — the variance of a Gaussian random process — should improve as $1/\sqrt{N}$, with $N$ the number of independent measurements. If the axion has expected linewidth $\delta_a$, then $N = t_\mathrm{av}\delta_a$. The averaging time at cavity resonance thus scales as $t_\mathrm{av} \propto 1/(\sigma(0)^2\delta_a)$. To estimate the appropriate size of the frequency step, I should find the detuning of an axion signal from cavity resonance that would double the time need to resolve it. More appropriate for the quasicontinuous tuning used in axion searches [36,37], this bandwidth $B$ is $B = \int_0^\infty \mathrm{d}\omega\,\sigma^2(\omega)/\sigma^2(0)$. The scan rate is thus

$$R(G,\kappa_\mathrm{m}) = \frac{B}{t_\mathrm{av}} = \left(\frac{2n_\mathrm{ax}^2\kappa_\mathrm{ax}^2\delta_a}{(\bar{n}_T + 1/2)^2}\right)\left(\frac{\sqrt{G}\kappa_\mathrm{m}^2}{\left[\kappa_\mathrm{loss}\kappa_\mathrm{m} + \frac{(\kappa_\mathrm{loss}-\kappa_\mathrm{m})^2}{4G}\right]^{3/2}}\right).$$ (13)

This rate can be maximized over both $G$ and $\kappa_\mathrm{m}$. If squeezing is not used ($G = 1$), maximizing the scan rate over $\kappa_\mathrm{m}$ has the interesting consequence that critical coupling is not the optimal value for $\kappa_\mathrm{m}$, but rather it is preferable to trade some reduction of on resonance sensitivity $\sigma(0)$ for wider bandwidth with $\kappa_\mathrm{m} = 2\kappa_\mathrm{loss}$ [47]. Turning on squeezing makes this trade more favorable with the optimum value of $\kappa_\mathrm{m} = 2G\kappa_\mathrm{loss}$ in the limit of large squeezing. Finally, in the limit of large squeezing, the ratio of the optimum scan rate with and without squeezing is just $G$. This increased scan rate is the quantum advantage of squeezing.

## 7 Conclusion

I conclude by noting that the technical challenges associated with implementing a quantum squeezed receiver within an axion search have been overcome. In particular, Josephson parametric amplifiers (JPAs) are now a routine piece of quantum technology [9,44,48,49], which prepare one and two-mode squeezed states of microwave resonators and measure (by amplification) single microwave quadratures. Recently two JPAs have been combined to realize the configuration in Fig. 4, creating a factor of two increase in axion scan rate over the CSL value [5]. The primary technical imperfection that limits the improvement to a factor of two is loss in transporting the squeezed states from their source to the cavity and then to the amplifier. The challenge is to do much better.

The mass of the axion has many decades of possible values still to be searched. And the signal is so feeble that at the CSL it would require existing axion search experiments many thousands of years to scan even one decade of possible masses at the more pessimistic model of axion-photon coupling [13]. Breaking through the quantum-limit is a critical step in improving this situation. If experiments were bound by the CSL, technical improvements that reduce noise and loss would have diminishing returns. Instead, a good idea for reducing loss can have a major impact on the axion search rate.

# Acknowledgements

My interest in quantum enhanced measurements in axion searches arose from my participation in the HAYSTAC search collaboration and from discussions that I had with Steve Girvin while I was visiting Yale. From Joe Kerckhoff, I learned systematic methods of solving quantum optics problems and Ben Brubaker taught me about the QCD axion theory and other ultra-light dark matter models. From the HAYSTAC team, particularly Maxime Malnou, Dan Palken, and Kelly Backes, I learned the detailed challenges of beating the quantum limit in a real axion search.

**Funding information** This work was supported by the DOE QUANTISED program, by the National Science Foundation, under Grants No. PHY1607223 and No. PHY1734006, and by Q-SEnSE: Quantum Systems through Entangled Science and Engineering (NSF QLCI Award OMA-2016244).

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
