# Peer review of "Quantum enhanced metrology in the search for fundamental physical phenomena"

_SciPost Physics Lecture Notes, doi:SciPost Phys. Lect. Notes 40 (2022)_

## Round 1 · Referee Report · Ivan Takmakov · 2021-11-1

Strengths

1. A nice theoretical introduction starting from the basic building blocks and assembling them into the full picture in the end.

2. While focusing on general principles and theory, this document also educates on experimental requirements and limitations.

Report

The lecture notes of Prof. Lehnert are a great pedagogical review on using the singe- and two-mode-squeezing techniques to speed up the currently ongoing microwave-frequency axion search. The text is well written, should be easy to follow for the target audience, and will be a great addition to the Les Houches lectures book. Some optional suggestions are listed in the "Required changes" section.

Requested changes

1. In Figure 2, a $\kappa$ label should probably be added, similar to the g label.

2. In Section 3.1:
- It would have been helpful if before diving into the excellent theoretical explanation, there was a short paragraph explaining a basic idea ( sort of an “abstract”) of the experiment. Then a reader who has some knowledge about circuit quantum electrodynamics could right away get some general intuitive picture and be able to see the motivation behind the few first sections. This might also be be useful for people with more 'experiment-oriented' mindset.

- It would have been helpful if it was explicitly mentioned that $\kappa$ is the TOTAL decay rate, accounting for both $Q_c$ and $Q_i$. Usually in cQED $\kappa$ is used for the $Q_c$ coupling, and it took me a bit to realize the context of the notes.

- Maybe consider rephrasing/changing paragraph 4 (“Notice that the dark matter…”). The message is clear, but the currently provided clarifications seem to be a bit confusing.

3. In Figure 3, mean(X) seems to be the same for top and bottom images for (B), but not for (C).

4. On page 8, it might make sense to redefine the “squeezing gain” as a “squeezing factor.” The word 'gain' originates from the parametric amplification, which is a source of the squeezed light/states. However, the generation is not the focus of these notes, and the 'gain' definition might confuse a reader.

5. On page 8, in the sentence “Imagine that I introduce a second oscillatory system (a second cavity in our example) that is uncoupled to the classical force we wish to estimate” it could add more clarity if the last part is changed to something like “that is coupled to the science cavity, but decoupled from the classical force we wish to estimate.”

6. In the end of section 4.2, as a reader I would be happy about having a short paragraph (or a relevant citation) comparing single- and two-mode squeezing and describing their pros and cons. Because, unlike in some other cases (e.g. Luke C G Govia and Aashish A Clerk, New J. Phys. (2017)), the two-mode squeezing doesn't seem to be a must-have for the axion detection. And, despite being physically beautiful and providing benefits (e.g. 2 measured quadratures, hence bigger information flux), it also comes with a price (more susceptible to cable losses, requires more complicated setup and analysis).

7. In Sec.6:
- For Ref.[38], the arxiv link seems to be wrong.
- On top of the nice and pedagogical Yurke’s paper [39] published in 1989, for the “JPAs are currently the best available technology” information it might be helpful for a reader to get some state-of-the-art citations.

8) In Sec.6.1, after the phrase “with a characteristic width $\delta_a \approx \omega_a/10^6$”, it would be nice to also mention the typical cavity linewidth so that a reader does not need to search for cavity quality factors in previous sections.

---

## Round 2 · Author Response

Thank you to the referee for his or her thoughtful comments. Changes have been made essentially as requested, with the exception of referee point 2. I have added an "abstract" paragraph in the new version, but at the end of section 2 rather in than in section 3.1. I thought that this addition was consistent with the intent of the referee's comment, but it seemed better organized to include the paragraph even earlier.

---

## Round 2 · List of Changes

1. Figure 2a now includes a \kappa annotation.

2. An abstract-like paragraph has been added to the end of section 2 to summarize the dark matter haloscope experiment for readers familiar with circuit quantum electrodynamics.

A. When first introduced in section 3.1, \kappa is now described as the total science cavity decay rate.

B. Paragraph 4 in section 3.1 has been rewritten to have a clearer topic sentence. The first sentence now reads "In this approximation, the quantum fluctuations of the dark matter field have no influence on the cavity’s evolution."

3. Figure 3 has been corrected so that the mean value of the two X distributions coincide, as intended.

4. On page 9, when G is first introduced it is now described as the "squeezing factor" rather than the "squeezer gain," but with a parenthetical stating that it is also known as the "squeezer gain."

5.n page 10, when introducing the notion of an ancilla cavity the ettesxt txt iads "Imagine that I introduce a second oscillatory system (a second cavity in our example) whose quantum state can be manipulated, but can be otherwise much simpler than the science cavity as it need not couple to the dark matter field." Hopefully, this phrasing provides better emphasis.

.6I nIthe end of section 4.2, I mention that in this application two-mode squeezing, though conceptually appealing, provides no benefit to an axion search over single-mode squeezing.

7. The arxiv link for (new manuscript) reference 41 is now correct. Citations to more recent JPA development papers are given when introducing Josephson parametric amplifiers in section 6, rather than just the 1989 Yurke paper.

8.) In section 6.1 the reader is reminded of the expected ratio of the axion to science cavity linewidth in a parenthetical after equation 10.

Other small changes:

A short footnote is added to the end of the first paragraph of section 6.2, where I mention the ability to squeeze thermal noise. In the footnote, I mention that a quantum limited amplifier alone, without squeezing, can overcome large thermal noise. In the context of dark matter searches, this case is considered in detail in the manuscript, arxiv:1803.01627, I reference I now include.

Minor typographical errors have also been corrected.

You are currently on this page

Resubmission 2110.04912v2 on 30 November 2021

---

## Editorial Decision

published